# Expression and Variation of the Genes Involved in Rhizobium Nodulation in Red Clover

**DOI:** 10.3390/plants11212888

**Published:** 2022-10-28

**Authors:** Randy D. Dinkins, Julie A. Hancock, Derek M. Bickhart, Michael L. Sullivan, Hongyan Zhu

**Affiliations:** 1Forage-Animal Production Research Unit, USDA-ARS, Lexington, KY 40506, USA; 2College of Agriculture, Food and Environment, University of Kentucky, Lexington, KY 40508, USA; 3US Dairy Forage Research Center, USDA-ARS, Madison, WI 53706, USA; 4Department of Plant and Soil Sciences, University of Kentucky, Lexington, KY 40546, USA

**Keywords:** forage legumes, nitrogen fixation, gene expression, nodulation, nodule-specific cysteine rich peptides

## Abstract

Red clover (*Trifolium pratense* L.) is an important forage crop and serves as a major contributor of nitrogen input in pasture settings because of its ability to fix atmospheric nitrogen. During the legume-rhizobial symbiosis, the host plant undergoes a large number of gene expression changes, leading to development of root nodules that house the rhizobium bacteria as they are converted into nitrogen-fixing bacteroids. Many of the genes involved in symbiosis are conserved across legume species, while others are species-specific with little or no homology across species and likely regulate the specific plant genotype/symbiont strain interactions. Red clover has not been widely used for studying symbiotic nitrogen fixation, primarily due to its outcrossing nature, making genetic analysis rather complicated. With the addition of recent annotated genomic resources and use of RNA-seq tools, we annotated and characterized a number of genes that are expressed only in nodule forming roots. These genes include those encoding nodule-specific cysteine rich peptides (NCRs) and nodule-specific Polycystin-1, Lipoxygenase, Alpha toxic (PLAT) domain proteins (NPDs). Our results show that red clover encodes one of the highest number of NCRs and ATS3-like/NPDs, which are postulated to increase nitrogen fixation efficiency, in the Inverted-Repeat Lacking Clade (IRLC) of legumes. Knowledge of the variation and expression of these genes in red clover will provide more insights into the function of these genes in regulating legume-rhizobial symbiosis and aid in breeding of red clover genotypes with increased nitrogen fixation efficiency.

## 1. Introduction

The root nodule symbiosis between legumes and nitrogen-fixing soil bacteria is vital for the supply of biologically fixed nitrogen to soil and plants in agricultural systems. Development of nitrogen-fixing nodules involves simultaneous differentiation of both symbiotic partners. During this process, the bacteria become adjusted to an endosymbiotic lifestyle and develop into mature bacteroids that are capable of nitrogen fixation. Depending on the host, the morphology and physiology of bacteroids can be strikingly different. In *phaseoloid* legumes (e.g., soybeans and common beans), the bacteroids mostly retain the same morphology and can revert to the free-living form. In contrast, galegoid legumes (e.g., alfalfa, clovers, and peas), the bacteria often undergo terminal differentiation, which is characterized by cell enlargement coupled with genome amplification, increased membrane permeability, and loss of reproductive ability [1,2]. It was reported that terminal differentiation may lead to more efficient nitrogen fixation compared to non-terminal differentiation [3]. The terminal differentiation of bacteroids in *Medicago* and related legumes is associated with the expression of nodule-specific peptides that are targeted to bacteroids in the symbiosome, which include nodule-specific cysteine-rich peptides (NCRs) [4], proline-rich peptides (PRPs) [5], calmodulin-like proteins (CaMLs) [6], glycine-rich proteins (GRPs) [7], LEED..PEED signature proteins (LPs) [8], and nodule-specific PLAT (Polycystin-1, Lipoxygenase, Alpha-Toxin) domain proteins (NPDs) [9,10]. Some of these peptides are essential for development of high-efficiency nitrogen-fixing nodules [4,11], while others are toxic to specific bacterial strains and thus lead to the development of nodules incapable of fixing nitrogen [11,12]. Some of these peptides appear to be species-specific, such as the LPs that have only been identified in Medicago to date [8], whereas others appear to be widespread yet have very little sequence homology between species and highly differentially expanded in the different species [13]. An example of the latter are the NCR peptides that are only present in inverted-repeat lacking chloroplast (IRLC) galegoid legumes and absent in *phaseoloid* legumes. It has been proposed that these peptides optimize nitrogen fixation performance as different legume species contain similar, but different, sets of NCR gene products that target different rhizobial species [14,15].

Clovers and other forage legumes are commonly utilized in pastures as they offer an opportunity to reduce nitrogen fertilizer use when inter-seeded with forage grasses [16]. Perennial clovers in a dense grass-clover pasture can contribute more than 200 kg nitrogen per ha/year to the cropping system [17]. Red clover is an ideal legume pasture choice for many areas of the world in that it tolerates soil acidity and poor drainage and is well suited for hay production and to grazing when inter-seeded with common forage-grasses [18,19]. Molecular work on red clover has lagged behind many other cultivated crop species despite its relatively small genome [20], primarily due to the fact that it is an outcrossing species making genetic gains more difficult [21]. Genomic resources have been developed that now allow for molecular characterization and analysis [22,23,24,25,26]. The high heterozygosity in red clover, while complicating the assembly of contiguous chromosome molecules, has allowed for the identification of chromosomal regions and genes involved in various functions using single nucleotide polymorphic loci (SNPs) mapping [26,27,28,29,30], including identification of red clover genotypes differing in their nitrogen fixation potential [31]. However, still lacking is information on the molecular response and interaction of red clover with rhizobium that has been primarily done on model legumes such as soybeans (*Glycine max*), *Lotus japonicus*, and *Medicago truncatula*. For example, it has been postulated that red clover encodes roughly 425 NCRs, suggesting that this species contains one of the higher number of these peptides [9], although there is no information on the expression of the genes encoding these peptides in red clover due to the lack of annotation of the genes in the databases. Thus, the goal of the current work was to identify, and monitor expression of, genes in the interaction of red clover plants with rhizobium by comparing gene expression in roots of inoculated with uninoculated plants.

## 2. Results and Discussion

In order to characterize the red clover genes associated with nodulation, analysis was done on roots where plants were inoculated (Nod+) to those that were not inoculated (Nod−). The plants for the current experiment were derived from the Kenland variety and from crosses derived from a putative *ifs1* mutant generated by CRISPR/Cas9 [32]. As the knock-down of the TpIFS1 gene had no effect on nodulation, and gene expression analysis revealed no interaction between nodulation and the IFS1 knockdown (see Figure 5 in Dinkins et al. [32]), comparisons of differential gene expression between the Nod+ and Nod− roots were done across both WT and *ifs1/ifs1* genotypes (Figure 1a—dendrogram). Of the 43,827 red clover gene models annotated in the ARS1 genome assembly, 32,285 genes had at least one mapped read from the root-derived mRNA. For the current study, a minimum of five reads was used as a cutoff for comparisons between the Nod+ and Nod− root samples, resulting in the root transcriptome for the current analysis of 26,128 red clover gene models. Furthermore, for the statistical analysis of differential expression based on the four replicates (individual plants) for each Nod+ and Nod− RNA sample, the criteria used for designating significant differential expression between the Nod+ and Nod− roots was greater than two-fold expression difference, and −log10 *p*-value of 2.12 (corresponding to *p*-value of 0.003 and based on an FDR *p*-value 0.1). Using the above criteria, 1474 red clover genes were differentially expressed (DEGs) between the Nod+ and Nod− roots (Figure 1b—volcano plot). Of these, 512 genes were down-regulated in Nod+ roots and 962 genes were up-regulated in Nod+ roots (Appendix A). Of the 962 DEGs showing enhanced expression in Nod+ roots, 398 were associated with the nodule-specific cysteine rich (NCR) peptides and cysteine-rich defensin peptides (described in more detail below). The ten most highly expressed genes in the Nod+ root samples include those encoding four leghemoglobin Lb120-1 homologues, two nodulin-25 like proteins, an early nodulin-75-like protein, a nodulin-specific embryo expressed 3-like protein, a nodule-specific glycine-rich protein and a putative NMS32/34 nodulin-specific protein (Table 1). The high expression of these genes is similar to results of the most highly expressed genes found during nodulation in *Medicago* species (Burghardt et al., 2017; Huang et al., 2022). The expression of these genes were also some of the most highly differentially expressed genes when comparing the Nod+ and Nod− root samples (i.e., no or very low expression in Nod− roots), confirming that the high expression is due to the presence of the nodules in these samples. The list of Nod+ greater than Nod− DEGs is presented in the supplemental file (Appendix A).

Here, we highlight an example of genes or gene families that were differentially expressed between the Nod+ and Nod− roots, with a focus on the genes that are up-regulated by nodulation.

### 2.1. Up-Regulated Genes by Nodulation

#### 2.1.1. NOD25 and N22

*NOD25* orthologs in *Medicago truncatula* (*MtNOD25*), *Medicago sativa* (*MsNOD25*) and *Vicia faba* (*VfNOD28/32*) are expressed only in root nodules and encode a nodule-specific protein designated nodulin-25. Nodulin-25 homologs consist of different repetitive modules that are highly diverged between legume species but the N-terminal signal peptide (SP) and the C-terminus are highly conserved. The N-terminal SP is required and sufficient to target the protein to the. symbiosome of infected root nodule cells but its function in symbiosis development remains unknown [33]. In red clover, we identified two *NOD25* loci on chromosome 3 in the current genome assembly but they most likely represent two allelic (haplotype) versions of the same gene. Nevertheless, there are over twenty-five different putative nodulin-25 protein isoforms predicted via alternative splicing from the two *NOD25* genes. The two *NOD25* genes had very high expression in the Nod+ roots but no expression was detected in Nod− roots. The RNA-seq data support most, although not all, of the different splicing mRNA products (Appendix A), suggesting that these might be expressed at different stages during the symbiosome development not observed in the current study, or that these predicted alternative splicing products may not occur in these plants.

The *NOD25* locus is highly syntenic in galegoid legumes. Immediately adjacent to the *NOD25* gene is another nodule-specific gene called *MtN22* in *M. truncatula* and NMS32/34 in *M. sativa*. In red clover, the *MtN22* ortholog appears to be duplicated; there are three copies in one putative haplotype (*Tp3g123916883*, *Tp3g123916884*, and *Tp3g123916889*) and two copies (*Tp3g123918940* and *Tp3g123918941*) in another. All the five genes were expressed only in the Nod+ roots (Table 1). *Tp3g123916883*, *Tp3g123918940* and *Tp3g123918941* are postulated to produce three protein isoforms each via differential splicing (Appendix A). From the RNA-seq data, it is difficult to differentiate differential splicing due to sequence similarities that results in similar protein products (Appendix A). *MtN22* also produces three putative protein isoforms and has the highest similarity to the red clover *Tp3g123918940*. Interestingly, the only other MtN22-like protein that is currently predicted in other species are in *Cicer arietinum* where three genes are found. In all probability, this lack of the matches to other legumes, such as to the *Viciaea*, may be due to the lack of gene information at this time.

#### 2.1.2. NCRs

Of the 962 DEGs up-regulated in the Nod+ roots, 393 encode the nodule-specific cysteine rich (NCR) peptides. Many of the NCR genes in the AR-100 genome assembly were annotated as non-coding RNA (ncRNA) genes. Thus, we manually re-annotated the genes based on known conserved NCR sequences and gene structure. Specifically, these genes were identified by the presence of two (or sometimes three) exons, where the first exon primarily encodes the signal peptide and the second exon codes the mature peptide containing four or six cysteine residues as described further below. Where a ncRNA locus was associated with the NCR gene, the locus number was maintained, however in instances where no locus was identified, we manually annotated according to chromosome number and start location of the gene on the chromosome (Appendix A). Expression of the NCR genes varied in the Nod+ roots, but essentially no expression was observed in the uninoculated root samples (Appendix A).

The distinguishing feature of these small peptides are the 4–6 conserved cysteine residues [15]. The four cysteine peptides were conserved C-(X4)-DC-(X11-17)-C-(X4)-C, the six cysteine peptides were conserved C-(X3-6)-C-(X4-7)-C-(X4-7)-C-(X4-17)-C-(X1)-C (Appendix A). The similarity of the C-terminal CXC motif of the six-cysteine group with the cys-defensin family of proteins suggests that disulphide bridging is a component of the protein conformation for mode of action as an antimicrobial peptide.

A total of 460 putative NCR genes were identified in the assembly of which 425 were expressed in the current study. Genome location and expression of the putative NCR genes are listed in Supplemental File (Appendix A). The predominant gene structure of the NCR genes in red clover was two exons interrupted by an intron near the signal peptide cleavage site. However, 32 genes also contain a second intron at the 3′ end that included one to a few codons prior the stop codon. The inclusion of the third exon was annotated with the aid of 3′ Tag-sequencing data [34]. There were a few (18) that appear to be intronless. A predicted signal peptide was found for 425 putative NCR peptides using the SignalP 4.1 software in CLC Genomics Workbench, while the remaining 34 did not have a predicted signal peptide but had the conserved cysteine amino acids and are included as they were expressed in the current study.

The distribution of the number of NCR peptides based on the isoelectric point ranges is presented in Figure 2. It has been suggested that cationic peptides, those with the isoelectric point above 9.0, can function as antimicrobial peptides [15,35,36] of which the MtNCR247 has been specifically shown to be a strong antimicrobial peptide [35]. There are 149 expressed cationic NCR peptides in red clover which represents 32% of the red clover NCR peptides, while *M. truncatula* proportion is roughly 15%, alfalfa (*Medicago sativa*) is currently postulated to have 10%, and sweet clover (*Melilotus offiicinalis*) and pea (*Pisum sativum*), 9% and 8.5%, respectively. Interestingly, red clover only has a single anionic NCR with an isoelectric point less than 4.0, whereas the other species above contain roughly 6%.

There are 25 genes that encode proteins containing eight, or more, cysteine residues. These are not listed as NCR genes and likely encode putative defensins [37,38]. Nevertheless, these genes are also observed to be primarily expressed in the Nod+ roots, have an N-terminal signal peptide, and thus likely targeted to the symbiosome.

#### 2.1.3. ATS3-like/NPDs

Also observed in the highest expressed Nod+ gene list (Table 1) is a gene encoding a class of cysteine-rich peptides with homology to the *Arabidopsis thaliana* embryo-specific protein-3, or ATS3-like. In *M. truncatula* the nodule-specific ATS3-like genes are comprised of five closely related members (Appendix A), encoding proteins that have been named as nodule-specific Polycystin-1, Lipoxygenase, Alpha toxic (PLAT) domain proteins, or NPD’s [9]. In red clover this gene family has expanded to 12 genes that are all tightly linked on chromosome 7 (Appendix A). All of the red clover ATS3-like/NPD genes were expressed only in the Nod+ roots, most at high levels, including the *TpNPD5* (*Tp7g123899073*) that was one of the most highly expressed genes in this study (Table 1). Analysis using SignalP 4.1 software suggests that these proteins are cleaved and most likely targeted to the symbiosome as previously found for MtNPD1 [10]. The *TpNPD7* gene, while actively transcribed, probably encodes a non-function protein in the sequenced cultivar as it lacks a motif encoded by exon 2 and a small deletion in the C-terminus (Appendix A). Additionally, the *TpNDP9* locus, while also apparently transcribed, may be a pseudogene as the third exon is not proximal to the first two, although RNA-seq expression data suggests possible inclusion of the exon 3 sequence shared with *TpNPD12* (Appendix A). *TpNPD12* appears to encode two protein isoforms with a two amino acid difference due to alternative splicing in exon 2, both supported in the RNA-seq mapping results. The rationale for the expansion of this gene family in red clover is not known. In *M. truncatula* it appears that these proteins may function in the selection for, or against, different strains of rhizobium [10,39], and thus it will be of interest to characterize this gene family in red clover as to whether it could aid in identifying better interacting *R. leguminosarum* strains with different red clover genotypes.

#### 2.1.4. NAD1/NIN

Other genes involved in early nodulation include *NODULES WITH ACTIVATED DEFENSE 1* (*NAD1*) and *NODULE INCEPTION* (*NIN)*. MtNAD1 is essential for the control of plant defense during the colonization of the nitrogen-fixing nodule and is required for bacteroid persistence. NIN orthologs are required to initiate nodule formation in the cortex [40] and to transcriptionally activate NF-YA1 (Nuclear Factor Y), a transcription factor [41,42] involved in various nodulation steps [41,43,44]. In the epidermis, NIN is necessary for the onset of rhizobium infection [45,46] but restricts the expression of *EARLY NODULIN11* (*ENOD11*), a marker of the pre- infection and infection steps [47,48]. In red clover, the putative orthologs of *NAD1* (*Tp6g23891533*) and *NIN* (*Tp4g123881749*) both were up-regulated in the Nod+ roots. The *TpNIN* promoter region has the same upstream elements that regulate transcription as described for *MtNIN*, suggesting that regulation is probably similar [49].

The *MtNF**-YA1* that controls meristem function is repressed in nodules through interaction with miR169, which is mediated by a small 62-aa peptide encoded by an alternatively spliced 5′-end of the *MtNF**-YA1* gene that reduces expression [50]. The putative NF-YA1 ortholog in red clover (*Tp1g123902956*) was expressed as a significant DEG in the current study. Additionally, the alternative splicing of the first intron seen in *MtNF-YA1* was also observed in *TpNF-YA1*, suggesting a conserved regulation mechanism via miRNA169 [51].

#### 2.1.5. DNF1-like and SYP132

Other components of the secretory pathway that are targeted to the symbiosome in *M. truncatula* include components of the signal peptidase complex (SPC) that include *dnf1* (*defective in nitrogen fixation 1*). Putative red clover genes that could encode putative orthologues of *dnf1* are: *Tp6g123890890*, *Tp6g123917546* or *Tp1g123900039*. None of the three genes were observed to be expressed only in the Nod+ roots, although *Tp6g123917546* was observed to have a two-fold increase in expression in the Nod+ roots, although not declared significantly different. No expression differences between the Nod+ and Nod− roots were observed in the other putative orthologues of the other subunits to the signal peptidase complex (DAS12, DAS18 and DAS25) [52].

Another protein involved in the transport and protein targeting to the symbiosome is SYNTAXIN 132 (SYP132). In *M. truncatula*, the *SYP132* gene encodes two isoforms that distinguish host plasma membrane and symbiosome membrane via alternative splicing and polyadenylation [53,54]. While no significant differences in overall gene expression of the putative red clover *SYP132* (*Tp7g123897430*) between Nod+ and Nod− roots was observed, only reads from Nod+ root samples mapped to the alternative spliced isoform suggesting that *Tp7g123897430* gene is the red clover SYP132 ortholog. Complicating the analysis of the putative *TpSYP132* gene is the presence of an overlapping gene (*Tp7g123897432*) at the 3′ end with the alternatively spliced isoform (Appendix A). The *Tp7g123897432* gene is currently labeled as a ncRNA gene and is expressed in both Nod+ and Nod− roots. RNA-seq data suggests that this (*Tp7g123897432*) gene also produces alternative spliced mRNA isoforms but, as the ncRNA designations suggests, no open reading frames have been found. Additional analysis will be necessary to determine the function of this ncRNA and whether there might be any involvement in regulation of the red clover *SYP132* gene as suggested for *M. truncatula* [55].

#### 2.1.6. Transcription Factor-Encoding Genes

One of the most highly expressed transcription factor-encoding genes was *Tp6g123889481* that encodes a C_2_H_2_ zinc finger protein that is orthologous to the *M. truncatula* REGULATION OF SYMBIOSOME DEVELOPMENT (RSD) [56]. Another Nod+-specific C_2_H_2_ zinc finger gene, *Tp6g123890030* is orthologous *MtrunA17_Chr7g0231181*, which is also expressed only in nodules in *M. truncatula* [57]. The role of this putative C_2_H_2_
*LOC123890030* gene product in nodulation is still unknown.

### 2.2. Down-Regulated Genes by Nodulation

A list of DEGs that were down-regulated in the Nod+ roots is presented in Appendix A. These include a number of genes involved in pathogen response and biosynthesis of terpenoids (Table 2). This is consistent with the observations that the immune responses were suppressed during the symbiosis development and maintenance. Gene Ontology, Biological Process, Molecular Function and Cellular Component terms associated with these down-regulated genes are presented in Table 3. Expression of genes with molecular function GO:0016705 (oxidoreductase activity) was associated with a number of cytochrome P450′s. These genes are likely involved in terpenoid biosynthesis as previously highlighted as highly differentially expressed.

While not highlighted in the GO pathways, *Tp4g123921773*, a gene encoding a protein with homology to the LysM/CERK Nod Factor Receptors from *Lotus japonicus* (NFR5), soybeans (NFRa and NFRb) and pea (SYM37A) was also significantly down-regulated in the Nod− roots (Table 2). Other putative LysM/CERK homologs, *Tp4g123921715* and *Tp2g123904164*, showed a similar expression pattern. While these gene products have been shown to be the receptors for rhizobium and determinants that specify species-specific plant-rhizobium interactions [58] and it has not been demonstrated that these loci are the specific receptors in the interactions of red clover and rhizobia. Down-regulation of these genes in Nod+ roots may also be associated with suppression of immune responses because LysM-domain containing receptors also serve as immune receptors that perceive common microbe-associated molecular patterns such as chitin to trigger immunity signaling [59].

## 3. Materials and Methods

### 3.1. Plant Growth

Seeds of the Kenland cultivar and from crosses derived from a putative *isoflavone synthase 1* or *ifs1* mutant [32] were germinated on sterile moist filter paper and the germinated seedlings were individually transferred to 5-cm pots containing autoclaved sand. Seedlings were grown for two weeks in a growth chamber programmed for 16 h light at 22 °C and 8 h dark at 20 °C. The plants were then inoculated with a commercial strain of *Rhizobium leguminosarum*
*biovar trifolii* sold as N-Dure^TM^ True Clover (Verdesian, Cary, NC, USA). The un-inoculated plants were used as controls. Water was applied twice daily using an overhead sprinkler system for six weeks. Following six weeks of growth, the nodulated roots (including both nodules and roots, designated as Nod+) and uninoculated roots (Nod−) were collected for transcriptome analysis.

### 3.2. RNA Extraction and Library Preparation

RNA was isolated and processed according to methods described in Dinkins et al. [32] and the frozen total RNA samples were submitted to NovoGene (Seattle, WA, USA) for the construction of 150-base paired-end mRNA libraries and subsequent sequencing following the manufacture procedures (https://en.novogene.com/services/research-services/transcriptome-sequencing/eukaryote-mrna-sequencing/, accessed on November 28 2019). Sequencing was done using Illumina Hi-Seq sequencing, resulting in a minimum 6 GB of sequence data per library. The raw RNA-seq reads have been deposited in the Short Read Archive database at NCBI under BioProject PRJNA657870.

### 3.3. Analysis of RNA-Seq Reads

Raw reads were filtered to remove reads less than 30 bases and then mapped to the red clover genome assembly (version ARS_RC1.1; GenBank Accession No. GCA_020283565.1) [25] using the RNA-Seq tool, CLC Genomics Workbench (v22.0; CLC Genomics Workbench, Qiagen Aarhus A/S, Denmark). Minimum length fraction was set to 0.5 and minimum similarity fraction set to 0.8. Reads mapped to each gene were transformed by addition of one (1) to eliminate zeros and normalized by the Quantile method in CLC Genomics Workbench. Only genes that had a minimum of five average reads over either the Nod+ or Nod− samples were retained for differential expression analysis.

Manual annotation was performed for some of red clover genes based on the RNA-seq exon/intron expression profiles. This included manual annotation of the red clover NCR genes, many of which were annotated as ncRNAs in the current assembly, in order to identify the open reading frames. Thus we manually re-annotated the genes based on known conserved NCR gene structure and from sequences obtained from genes derived using the SPADA software [60] in the red clover genome assembly Tpv2.1 (GenBank Accession No. GCA_900079335.1). Following re-annotation, the RNA-seq tool in CLC Workbench was re-run to obtain the corrected read expression data. The SignalP 4.1 software in CLC Genomics Workbench [61,62] was used to predict the signal peptide cleavage site.

For differential expression analysis, the transformed read number in each sample was analyzed using JMP Genomics (version 9.0—SAS Institute, Cary, NC, USA). The normalized reads for each sample were log transformed (log2) using the JMP Basic RNA-Seq Workflow analysis of variance (ANOVA). Comparisons were done between Nod+ and Nod− samples over four replications (samples from four plants each). Based on the error estimated by ANOVA for each gene-by-gene comparison, significant differential expression was calculated in the comparisons between the *IFS1* wild-type and *ifs1* mutant expression means. A false discovery rate (FDR) multiple testing method at -log_10_
*p*-value of 2, that corresponds to an FDR *p*-value of 0.1, and a 2-fold difference were chosen as criteria for significance.

Pathway analysis of the differentially expressed genes was done using the annotation and gene ontology term from the *Trifolium pratense* Gene Ontology list in Legume Information Service (https://legumeinfo.org, accessed on 7 September 2021) using the web-based AgriGOv2 (http://systemsbiology.cau.edu.cn/agriGOv2/, accessed on 25 July 2022) [63].

## 4. Conclusions

Red clover has not been a major contributor in legume/rhizobium studies primarily due to the issues associated with self-incompatibility and difficulties obtaining homozygous plants for genetic analysis compared to the model species currently used. Using recently developed red clover genomic resources and RNA-seq expression analysis, we describe the genes that are expressed in nodule forming roots. As has been observed for other species, for red clover many of the genes involved in symbiosis are conserved with those of other legumes, while other genes are plant species-specific with little or no homology across species. These include the nodule-specific cysteine rich proteins (NCRs) that are found only in the IRLC legume clade, that while have common features, i.e., conserved cysteine residues, lack sequence homology and likely regulate the specific plant genotype/symbiont strain interactions. With the knowledge of the expression of these genes and genetic variation in red clover, breeding for increased nitrogen fixation efficiency by selection of genotypes with increased rhizobium specificity should be feasible.

## Figures and Tables

**Figure 1 plants-11-02888-f001:**
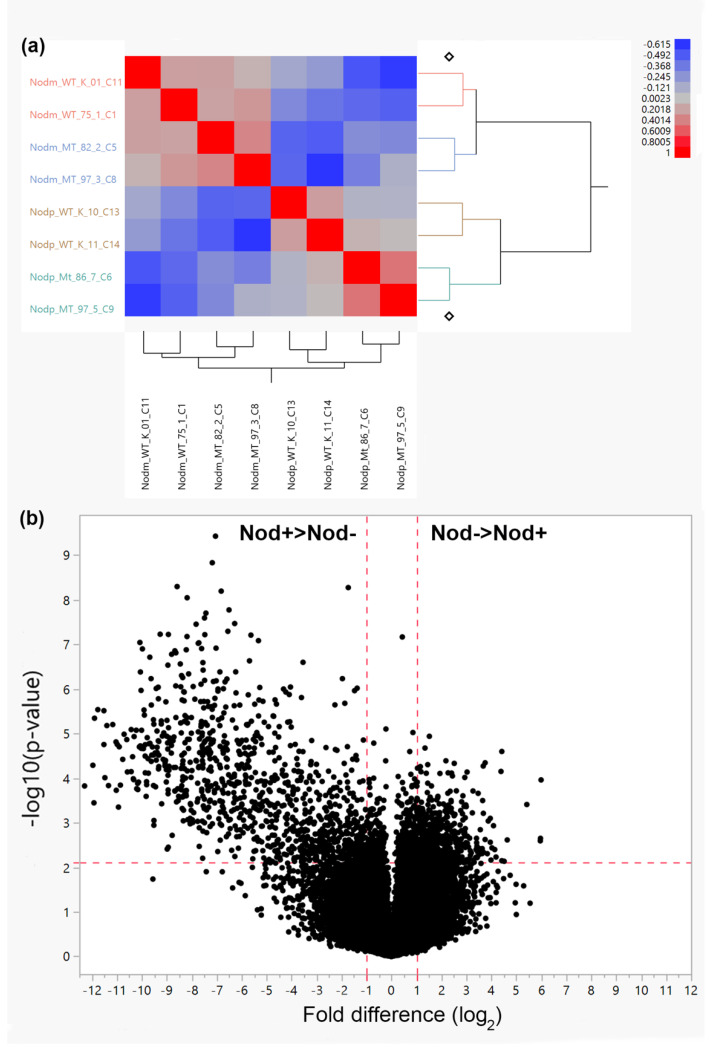
(**a**) Dendogram and heat map of Nod+ (labeled as Nodp for the samples displayed in the figure vertical axis) and Nod− (labeled as Nodm for the samples displayed in the vertical axis) expression profiles for each individual sample and (**b**) volcano plot showing differential expression between Nod+ (NodP) and Nod− (NodM) root samples.

**Figure 2 plants-11-02888-f002:**
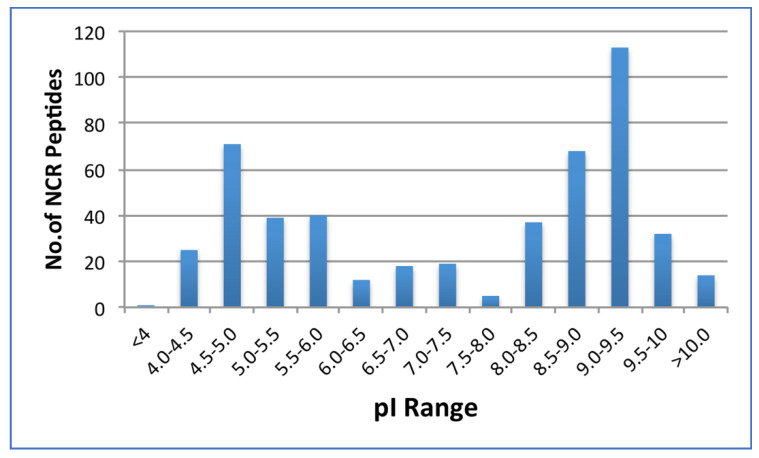
Number of the red clover NCR peptides distributed by isoelectric point (pI).

**Table 1 plants-11-02888-t001:** Highest expressing red clover genes in Nod+ root samples.

Feature ID	Normalized Reads (Means) Nod−	Normalized Reads (Means) Nod+	Log2 Diff Express (Nod+) /(Nod−)	Product
LOC123882175	381.52	74,205.38	7.60	leghemoglobin_Lb120-1
LOC123882181	31.62	63,699.05	10.98	leghemoglobin_Lb120-1
LOC123916815	15.55	57,593.44	11.86	nodulin-25-like isoform X10
LOC123902448	328.64	38,271.05	6.86	early nodulin-75-like
LOC123882180	7.62	28,608.07	11.88	leghemoglobin_Lb120-1
LOC123917408	97.65	27,409.22	8.13	nodulin-26-like isoform X2
LOC123882176	17.47	24,305.17	10.44	leghemoglobin_Lb120-1
LOC123899073	4.50	22,133.74	12.26	Embryo-specific protein ATS3B-like; NPD
LOC123884281	5.08	20,011.87	11.95	ctenidin-1-like
LOC123916883	6.72	17,961.34	11.38	NMS32/34 protein, putative
LOC123915658	3049.67	13,975.22	2.20	bifunctional aspartate aminotransferase
LOC123893140	4110.65	13,859.39	1.75	legume-specific_protein
LOC123918940	4.23	13,807.55	11.67	NMS32/34 protein, putative
LOC123891069	18.14	13,225.90	9.51	leghemoglobin_Lb120-1
LOC123888452	57.99	12,010.79	7.69	early_nodulin_ENOD18
LOC123913320	2502.68	11,384.90	2.19	glutamine synthetase nodule isozyme
LOC123909356	6.27	10,611.03	10.72	basic blue protein-like
LOC123895631	2916.89	10,575.69	1.86	inactive beta-amylase 9
LOC123921269	829.79	10,351.60	3.64	chaperone protein dnaJ 8, chloroplastic
LOC123911535	4108.67	10,330.98	1.33	heavy metal-associated isoprenylated plant protein 6-like

**Table 2 plants-11-02888-t002:** Most differentially expressed red clover genes in Nod− root samples compared to Nod+ samples.

Feature ID	Normalized Reads (Means) Nod−	Normalized Reads (Means) Nod+	Log2 Diff Express (Nod+) /(Nod−)	Product
LOC123899143	201.60	3.08	6.03	wound-responsive_family_protein
LOC123898241	210.61	3.43	5.94	wound-responsive_family_protein
LOC123899195	171.98	2.86	5.91	structural_constituent_of_cell_wall_protein,_putative
LOC123900972	110.72	2.63	5.39	sesquiterpene_synthase
LOC123923258	99.89	3.96	4.66	sesquiterpene_synthase
LOC123908919	312.53	13.93	4.49	geranylgeranyl_pyrophosphate_synthase
LOC123891221	66.37	3.09	4.42	disease_resistance_protein_(TIR-NBS-LRR_class)
LOC123920096	45.48	2.20	4.37	replication_factor-A_carboxy-terminal_domain_protein
LOC123917771	62.76	3.61	4.12	translation_elongation_factor_EF1B,_gamma_chain
LOC123908518	518.67	30.81	4.07	benzyl_alcohol_O-benzoyltransferase
LOC123924362	826.36	52.79	3.97	polygalacturonase_plant-like_protein
LOC123921773	1639.61	108.42	3.92	LysM_receptor_kinase_K1B
LOC123899196	98.73	6.92	3.83	1-aminocyclopropane-1-carboxylate_oxidase-like_protein
LOC123891479	241.74	18.20	3.73	cysteine/histidine-rich_C1_domain_protein
LOC123921461	230.24	17.49	3.72	cytochrome_P450_family_71_protein
LOC123912953	25.87	2.06	3.65	cytochrome_P450_family_71_protein
LOC123909630	39.54	3.18	3.64	amino acid transporter AVT1H
LOC123919510	15,190.12	1224.73	3.63	alpha-copaene synthase-like
LOC123891304	59.92	4.84	3.63	zinc finger protein 6-like
LOC123919667	210.91	17.06	3.63	disease_resistance-responsive,_dirigent_domain_protein

**Table 3 plants-11-02888-t003:** Gene ontology (GO) Biological Process (P), Molecular Function (F) and Cellular Component (C) terms terms associated with differentially expressed genes (DEGs) of red clover Nod− plants greater than Nod+ plants.

GO Term	Ontology	Description	Number in Input List	Number in BG/Ref	*p*-Value	FDR
GO:0071555	P	cell wall organization	8	62	6.10 × 10^−5^	0.016
GO:0045229	P	external encapsulating structure organization	8	62	6.10 × 10^−5^	0.016
GO:0071554	P	cell wall organization or biogenesis	8	84	5.10 × 10^−4^	0.077
GO:0042545	P	cell wall modification	6	48	0.0006	0.077
GO:0030599	F	pectinesterase activity	6	48	0.0006	0.059
GO:0020037	F	heme binding	20	384	0.00042	0.059
GO:0046906	F	tetrapyrrole binding	20	385	0.00044	0.059
GO:0052689	F	carboxylic ester hydrolase activity	7	68	0.00071	0.059
GO:0016705	F	oxidoreductase activity, acting on paired donors, with incorporation or reduction of molecular oxygen	16	292	0.00086	0.059
GO:0030312	C	external encapsulating structure	9	76	0.000043	0.0021
GO:0005618	C	cell wall	9	76	0.000043	0.0021

## Data Availability

The raw RNA-seq reads are deposited in the Short Read Archive database at GenBank under BioProject PRJNA657870.

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
