# Peer review of "Expression and Variation of the Genes Involved in Rhizobium Nodulation in Red Clover"

_plants, 2022, doi:10.3390/plants11212888_

Round 1
Reviewer 1 Report
This manuscript describes the results of a comparative RNAseq experiment in red clover in which samples of nodulated roots were compared with samples of un-nodulated roots. There are some interesting results described, notably the many NCR genes present in red clover. This gives clear pointers to further work that could be carried out to elucidate their possible role in specificity in the interaction between the two partners in the symbiosis.
I have found it difficult to find a lot wrong with this. My main query is about the experimental design. Two types of plants were used: The WT IFS1 and the mutant ifs1 (or crosses derived there off). Does that mean that 4 treatments were used: WT nod+, WT nod-, ifs1 nod+ and ifs nod-? It says that there are 4 reps of each treatment, but I can only see 4 nod+ and 4 nod- samples. Can you please clarify that. Also, it was not clear to me what exactly was the purpose of the ifs1 mutant (or derivative of) to be included in this work, other than perhaps confirm that the gene involved was expressed to a lesser degree in the mutant than in the wild type.
I found one type: Line 314 - change "japicius" to "japonicus".
Author Response
Reviewer Comment #1: This manuscript describes the results of a comparative RNAseq experiment in red clover in which samples of nodulated roots were compared with samples of un-nodulated roots. There are some interesting results described, notably the many NCR genes present in red clover. This gives clear pointers to further work that could be carried out to elucidate their possible role in specificity in the interaction between the two partners in the symbiosis.
Reply: Thank you for the positive comments on the objective of this work. We are very interested in furthering our work elucidating the roles of these genes, especially the NCR’s and NPD’s, in the red clover/ rhizobium symbiosis. However, we thought it would be helpful to elicit interest from others in the community in red clover / rhizobium symbiosis, and the reason we would like to make this work public.
Reviewer Comment #2: I have found it difficult to find a lot wrong with this. My main query is about the experimental design. Two types of plants were used: The WT IFS1 and the mutant ifs1 (or crosses derived there off). Does that mean that 4 treatments were used: WT nod+, WT nod-, ifs1 nod+ and ifs nod-? It says that there are 4 reps of each treatment, but I can only see 4 nod+ and 4 nod- samples. Can you please clarify that. Also, it was not clear to me what exactly was the purpose of the ifs1 mutant (or derivative of) to be included in this work, other than perhaps confirm that the gene involved was expressed to a lesser degree in the mutant than in the wild type.
Reply: We apologize for the confusion in using both wild-type and the CRISPR isoflavone synthase (ifs/ifs) knock-down lines for this study. As we state in the Materials and methods, the RNA-seq samples used in this study are the same plants previously described in Dinkins et al (2021) Plant Cell Rep. 40:517 (available at GenBank under BioProject PRJNA657870). As we pointed out, since there was no interaction in the expression of Nod+/Nod- DEGs and the IFS/IFS vs ifs/ifs DEG’s (shown in Figure 5A in the reference above), we combined the wild-type and ifs KO reads for the analysis of the Nod+ and Nod- samples. This allowed for analysis of four replications of the Nod+ and Nod- reads as the reviewer points out above. Since the result of the statistical analysis has previously been shown (i.e. published), we only referenced this result in the current manuscript. The expression profiles of the wild type and ifs/ifs KO plants can be seen to separate within each of the Nod+ and Nod- samples as seen in the phylogram presented in Figure 1a. Again, the results from the analysis of ifs/ifs KO have already been published and not part of the current analysis on the genes involved in nodulation. While the analysis can be (and was) done for the four treatments as suggested by the reviewer above, the lack of interaction does not change the expression differences observed in the Nod+ and Nod- plants, only the significance level used to declare DEGs statistically different.
Reviewer Comment #3: I found one type: Line 314 - change "japicius" to "japonicus".
Reply: Thank you for catching this. It has been corrected in the revised manuscript.
Reviewer 2 Report
this is interesting for clover breeders, I would have liked to have seen some qPCR, but in this case I don't think it's entirely necessary.
Author Response
Reviewer #2
Comments and Suggestions for Authors
Reviewer Comment #1: this is interesting for clover breeders, I would have liked to have seen some qPCR, but in this case I don't think it's entirely necessary.
Reply: Thank you for your positive response, and for stating that a second method showing expression results, i.e. qRT-PCR, should not be necessary. It is not uncommon to have requests for qRT-PCR data to corroborate RNA-seq data. We agree that this information is important when specific genes are being analyzed for phenotype and function, however for a “general picture” of overall expression, we feel that there is little added value. Numerous laboratories/publications, including ours, have consistently shown across multiples species, that RNA-seq, qRT-PCR and PolyA-Tag data, while they can result in different specific numerical values for individual genes, provide similar measures of gene expression. Each of the methods have their inherent strengths and weaknesses based on their underlying assumptions, however, we contend that RNA-seq is a robust method for determining overall genome expression as it is based on expression of long sequences (in our case, 150 base paired end reads) across each gene. Of course, this is predicated on the assumption that the library was constructed correctly, and the reads cover the exonic region of the annotated genes targeted. This was verified in CLC Genomic Workbench where many individual reads were mapped and visualized. In fact, as stated in the manuscript, this method was used in the manual annotation of a number of genes, including many NCR’s, that were not annotated (or not annotated correctly) in the current red clover genome assembly. Thus, while multiple gene expression methods can be used to provide the “best” measure of expression, each providing additional clarity, the costs and time associated with the additional experiments increases significantly for the returns provided.
Reviewer 3 Report
The manuscript is well written with lots of details and promising scientific results. The tables, tables, figures and supplemental data are organized neatly. I suggest to accept this manuscript.
Author Response
Reviewer #3
Comments and Suggestions for Authors
Reviewer Comment #1: The manuscript is well written with lots of details and promising scientific results. The tables, tables, figures and supplemental data are organized neatly. I suggest to accept this manuscript.
Reply: Thank you for the positive review.